# Serum Phytosterols Are Not Associated with Inflammatory Markers in Two Cross-Sectional, Swiss Population-Based Studies (The CoLaus|PsyCoLaus Study)

**DOI:** 10.3390/nu14122500

**Published:** 2022-06-16

**Authors:** Laura Stanasila, Pedro Marques-Vidal

**Affiliations:** Department of Medicine, Internal Medicine, Lausanne University Hospital, University of Lausanne, 46 Rue du Bugnon, 1011 Lausanne, Switzerland; laura.stanasila@unil.ch

**Keywords:** phytosterols, inflammation, epidemiology

## Abstract

Background: The association between inflammation and dietary sterols remains poorly assessed at the population level. Aims: To assess the possible association between serum levels of various phytosterols (PS) and inflammatory markers. Methods: Serum levels of six PS (campesterol, campestanol, stigmasterol, sitosterol, sitostanol, brassicasterol), four cholesterol synthesis markers (lathosterol, lanosterol, desmosterol, dihydroxylanosterol) and one cholesterol absorption marker (cholestanol) were measured together with levels of CRP, IL-6 and TNF-α in two cross-sectional surveys of a population-based, prospective study. Results: CRP levels were negatively associated with levels of cholestanol and of sterols of plant origin, although some associations were not statistically significant. CRP levels were positively associated with cholesterol synthesis markers in the first but not in the second follow-up. IL-6 levels were negatively associated with cholestanol in both follow-ups. No associations between IL-6 levels and PS were found in the first follow-up, while significant negative associations with campesterol, sitosterol, brassicasterol, sitostanol and campesterol:TC ratio were found in the second follow-up. TNF-α levels were negatively associated with cholestanol in both follow-ups. These associations did not withstand adjusting for sex, age, BMI and statin administration. Conclusions: In a population-based study, PS serum levels were not significantly associated with inflammatory markers.

## 1. Introduction

Cardiovascular disease (CVD) remains the leading cause of death in Western societies and worldwide [1,2]. Plant-based diets were shown to be beneficial with regard to cardiovascular, metabolic and mental health by lowering body mass index (BMI), blood pressure and inflammatory markers [3,4,5,6]. Dietary modification to incorporate larger amounts of plant foods, while simultaneously lowering the intake of animal-sourced foods, can be a potent tool in CVD prevention [7,8]. Inflammation is supposed to be one of the mechanisms at the core of CVD pathophysiology [9]. In recent decades, interest has developed regarding the anti-inflammatory action of plant-based diets [10,11,12,13]. While multiple mechanisms of action are likely at play regarding the anti-inflammatory effect of plant-based diets, specific bioactive nutrients found in plants are known to stand out as particularly effective per se, such as fiber, flavonoids, omega-3 fatty acids or phytosterols [14].

Phytosterols (PS) are naturally occurring cholesterol-like substances belonging to the triterpene family and are present in the cell membrane of most plant cells. They are commonly found in highest amounts in vegetable oils, nuts and oily seeds but can also be found in cereals and legumes, fruit and vegetables being the poorest food sources [15,16]. Sitosterol and campesterol (representing 56–79% and 18% of total PS, respectively) are two of the most abundant PS and therefore the most studied [17]. The estimated intake of phytosterols in Western diets oscillates between 78 and 358 mg/day [18], with cereal and oils providing an estimated 61% of the total PS intake.

Evidence is scarce with regard to a supposed anti-inflammatory effect of PS [19]. Long-term supplementation with PS decreased proinflammatory cytokine levels in apoE KO mice [20]. Although positive results were obtained in rodent studies [21,22,23,24], their transposal to humans has been inconsistent so far. Some studies reported encouraging findings. Beta-sitosterol was associated with decreased interleukin 6 (IL-6) and TNF-alpha (TNF-α) levels in both diabetic and non-diabetic subjects, confirming the results of animal studies [25]. PS supplementation improved TNF-α but not CRP nor IL-6 levels in patients with nonalcoholic fatty liver disease [26]. PS supplementation attenuated inflammatory pathways in healthy subjects in a proteomics study [27]. On the other hand, a meta-analysis of 20 randomized controlled trials found no significant effect of PS supplementation on inflammatory biomarkers (mainly CRP) in obese patients [28], and a more recent review also reported contradictory findings [29]. Furthermore, most studies relied on supplementation, while population-based studies of PS effects at the usual nutritional intake levels are scarce. Serum levels of PS were seldom quantified, and correlations were drawn most often between intake levels and measured endpoints. The majority of the studies focused on sitosterol and campesterol alone, without taking into account other molecules of the PS family that can also exert an effect.

Hence, our objective was to assess the association between serum levels of various phytosterols and inflammatory markers using data from two cross-sectional evaluations of a population-based cohort.

## 2. Materials and Methods

### 2.1. Population

The CoLaus|PsyCoLaus (www.colaus-psycolaus.ch) is a prospective cohort study established in 2003, following a sample of the inhabitants of the city of Lausanne (Switzerland), aged 35 to 75 years at baseline every 5 years [30]. In each survey, participants answered questionnaires, underwent clinical examination, and blood samples were drawn for analyses. Recruitment began in June 2003 and ended in May 2006; the first follow-up (FU) was performed between April 2009 and September 2012, and the second follow-up was performed between May 2014 and April 2017. Sterol assessment was performed in the first and the second follow-up.

### 2.2. Sterol Assessment

In both follow-ups, we measured serum concentrations for eleven sterols; one was a marker of cholesterol absorption (cholestanol); four were markers of cholesterol synthesis (lathosterol, lanosterol, desmosterol and dihydroxylanosterol); and six were sterols of plant origin (campesterol, campestanol, stigmasterol, sitosterol, sitostanol and brassicasterol).

Serum PS were assessed in the Department of Clinical Pharmacology, University of Bonn, Germany, as indicated previously [31]. Briefly, PS were extracted with cyclohexane before being separated with gas chromatography–mass-spectrometry-selected ion monitoring on a DB-XLB column (J&W Scientific, Folsom, CA, USA) using an HP-5890 Series II plus gas chromatograph combined with an HP-5972 mass selective detector (Hewlett-Packard, Böbligen, Germany).

Several ratios were computed from the data, as they were considered adequate markers of phytosterol intake: campesterol to cholestanol [32]; campesterol to TC; sitosterol to TC; and 5-α-cholestanol to TC [33].

### 2.3. Inflammatory Markers

Venous blood samples (50 mL) were drawn in the fasting state and allowed to clot. Serum was preferred to plasma, as it has been shown that different anticoagulants may affect absolute cytokine levels differently [34]. High-sensitive CRP (hs-CRP) was assessed with immunoassay and latex HS (IMMULITE 1000–High, Diagnostic Products Corporation, Los Angeles, CA, USA) with maximum intra- and inter-batch coefficients of variation of 1.3% and 4.6%, respectively. Serum samples were kept at −80 °C before assessment of IL-6 and TNF and sent in dry ice to the laboratory. Levels of these cytokines were measured using a multiplexed particle-based flow cytometric cytokine assay [35]. This methodology yields cytokine concentrations that correlate well with those obtained by other methods, such as ELISA [36]. Milliplex kits were purchased from Millipore (Zug, Switzerland). The procedures closely followed the manufacturer’s instructions. The analysis was conducted using a conventional flow cytometer (FC500 MPL, BeckmanCoulter, Nyon, Switzerland). Lower limits of detection for IL-6 and TNF-α were 0.2 pg/mL. A good agreement between signal and cytokine was found within the assay range (R^2^ ≥ 0.99). Intra and inter-assay coefficients of variation were, respectively, 16.9% and 16.1% for IL-6 and 12.5% and 13.5% for TNF-α.

### 2.4. Other Covariates

Smoking status was self-reported and categorized as never, former, current. Participants reported prescribed and over-the-counter drugs they were currently taking; statins, ezetimibe and metformin were considered due to their effect on cholesterol metabolism and/or inflammation. Diabetes was considered if the participants had a fasting plasma glucose ≥ 7.0 mmol/L or were taking antidiabetic drugs.

Body weight and height were measured with participants barefoot and in light indoor clothes. Body weight was measured in kilograms to the nearest 100 g using a Seca^®^ scale (Hamburg, Germany). Height was measured to the nearest 5 mm using a Seca^®^ (Hamburg, Germany) height gauge. Body mass index (BMI) was computed and categorized as normal (BMI < 25 kg/m^2^), overweight (25–29.9 kg/m^2^) and obese (30+ kg/m^2^).

### 2.5. Statistical Analysis

Statistical analyses were conducted using Stata v.16.1 (Stata Corp, College Station, TX, USA) separately for each survey. Results were expressed as number of participants (percentage) for categorical variables and as average (±standard deviation) or median [interquartile range] for continuous variables. Bivariate associations were computed using Spearman nonparametric test. Multivariable analyses were conducted using robust regression, and results were expressed as multivariable-adjusted slope and 95% confidence interval (CI). For multivariable analyses, adjustments were performed on age (continuous), sex (male, female), BMI (continuous), diabetes (yes, no) and statin use (yes, no). A second analysis was conducted replacing diabetes by metformin use (yes, no). Statistical significance was considered for a two-sided test with *p* < 0.05.

### 2.6. Ethical Statement

The institutional Ethics Committee of the University of Lausanne, which afterward became the Ethics Commission of Canton Vaud (www.cer-vd.ch), approved the baseline CoLaus study. The approval was renewed for the first and the second follow-up. The study was performed in agreement with the Helsinki Declaration and its former amendments, and in accordance with the applicable Swiss legislation. All participants gave their signed informed consent before entering the study.

## 3. Results

### 3.1. Sample Characteristics

The samples from the two follow-ups are described in Table 1. They consist of 730 and 526 subjects, respectively, with a women-to-men ratio slightly in favor of women. The average BMI reflected a majority of overweight and obese subjects. Almost half of the subjects never smoked, and a significant percentage received cholesterol-lowering medication, mostly statins.

### 3.2. Sterol Levels

The values of the 11 sterols from the two follow-ups are described in Table 2. Of the markers of cholesterol synthesis, lathosterol was the more abundant, followed by desmosterol, lanosterol and dihydroxylanosterol. Among the sterols of plant origin, the most abundant was campesterol, followed by sitosterol, brassicasterol, sitostanol campestanol and stigmasterol (Table 2). PS serum levels were found to be consistent from one follow-up to the other.

### 3.3. Association with Inflammatory Markers—Bivariate Analysis

Figure 1 displays the bivariate correlation matrix between CRP, IL-6 and TNF-α and serum sterol levels and ratios for the first (first matrix) and the second (second matrix) follow-up. The values of the correlation coefficients and their *p*-values are summarized in Appendix A. Overall, the patterns of positive and negative associations mirror each other to a high extent, showing good agreement between the two follow-ups. A significant and positive association was seen between levels of different PS and between their cholesterol-normalized levels. Cholesterol synthesis markers were mostly negatively associated with plant sterols and their ratios to total cholesterol (TC), especially in FU2. The cholestanol:TC ratio displayed a significantly negative association with cholesterol synthesis markers (lathosterol excepted) and with the cholesterol-normalized global level of cholesterol synthesis markers. Consistent positive associations were seen between CRP, IL-6 and TNF-α.

CRP levels were negatively associated with levels of cholestanol and of sterols of plant origin, although some associations were not statistically significant. CRP levels were positively associated with most cholesterol synthesis markers in FU1 but not in FU2 (Appendix A).

IL-6 levels were negatively associated with cholestanol in both follow-ups. No associations between IL-6 levels and sterols of plant origin were found in FU1, while significantly negative associations with campesterol, sitosterol, brassicasterol, sitostanol and campesterol:TC ratio were found in FU2 (Appendix A).

TNF-α levels were negatively associated with cholestanol in both follow-ups and with campesterol, sitostanol and brassicasterol in FU2 (Appendix A).

### 3.4. Association with Inflammatory Markers—Multivariable Analysis

The results of the multivariable analysis are summarized in Table 3. Most associations between phytosterol levels and inflammatory markers became non-significant. The only consistent associations (i.e., observed for both follow-ups) were found between CRP and synthesis markers-to-TC ratio and between TNF-α and cholestanol levels. Other inconsistent associations were found between CRP and lathosterol (positive), lanosterol (positive) and dihydro-lanosterol (positive); IL-6 and cholestanol (negative), dihydro-lanosterol (positive), stigmasterol (positive), cholestanol-to-TC ratio (negative), stigmasterol-to-TC ratio (positive) and phytosterols-to-TC ratio (positive); and TNF-α and campesterol (negative), cholestanol-to-TC ratio (negative), campesterol-to-TC ratio (negative) and sistosterol-to-TC ratio (negative) (Table 3). Similar findings were obtained when diabetes was replaced by metformin (Appendix A).

## 4. Discussion

Negative associations between CRP, IL-6 and TNF-α and sterols of vegetal origin were found in bivariate analysis, but those associations were no longer significant after adjusting for confounders. Overall, our results suggest that the effect of PS on inflammation is nonexistent or very limited.

### 4.1. Sterol Levels

PS serum levels in our study were very close to those reported in German volunteers [37], probably reflecting similar dietary patterns in the two populations. Sitosterol and campesterol levels in our study were also comparable to those reported in the Adventist Health Study-2 cohort [33]. Conversely, the phytosterol levels in our study were much lower than those reported in an Amish population [38]. A possible explanation is that the consumption of vegetables among CoLaus participants was particularly low, only 7% of them complying with the Swiss dietary guidelines [39]. Interestingly, a US study found no significant differences in phytosterol serum levels between vegetarians and non-vegetarians, despite a 60% difference in vegetable dietary intake [33]. The findings from this US study suggest counter-intuitively that there is no clear-cut association between dietary intake and blood levels of PS. Indeed, both cholesterol and PS absorption have been shown to be genetically determined [38,40,41,42]. High cholesterol absorption is associated with risk alleles in ABCG8 and ABO and with CVD [41], and a recent genome-wide association scan identified five other loci implicated in PS absorption [40]. Overall, it is possible that the association between PS and CVD might be due to the genetic background of the participants (i.e., higher or lower digestive absorption) rather than to the dietary intake of PS. The differing genetic background might also partly explain the differences in PS levels between populations. Overall, our results are comparable to those obtained in other countries and help establish the reference levels for PS in European populations.

### 4.2. Association between Sterol Levels and Inflammatory Markers

Negative associations between CPR, cholestanol and PS levels were found in both follow-ups. Negative associations between IL-6 and TNF-α with cholestanol and PS were found in FU2 but not in FU1. Our findings are partly in agreement with a study conducted in Japan, where negative associations were found between IL-6 and TNF-α with sitosterol but not with campesterol [25]. In the same study, the associations resisted multivariate adjustment, but the authors used a stepwise multiple regression analysis, and it is unclear if the associations were adjusted for all confounders. In our study, most associations became nonsignificant or inconsistent (i.e., observed in one study period but not in the other) after multivariate adjustment, suggesting that the effect of serum PS on inflammatory markers is small. This finding is in agreement with the conclusion of several systematic reviews and a meta-analysis, showing no significant effect of PS supplementation on inflammatory markers [28,29].

### 4.3. Implication for Public Health

In our community-dwelling, population-based study, the dietary intake of PS was insufficient to influence inflammatory markers. The Adventist Health Study-2 reported an average total PS intake below 450 mg/day [43]. This value is four times lower than those used in most randomized controlled trials [29], which failed to report any effect on inflammatory markers. Hence, our results suggest that increasing PS intake, either via dietary changes or via supplementation, will not lead to clinically relevant changes in inflammatory markers.

### 4.4. Strengths and Limitations

One of the strengths of this study is that it measured a large number of different PS and cholesterol absorption and synthesis markers in a large sample of a community-dwelling population. Furthermore, two different follow-ups were performed, which enabled replication of the findings in terms of the absolute and cholesterol-standardized PS serum levels, thus allowing for an internal validation of the procedure. To our knowledge, this is one of few studies reporting serum concentration levels for a wide array of PS under normal dietary conditions. Most reports so far came from randomized controlled trials that assessed PS levels in a small number of subjects [27,44,45] or that focused on a small number of PS, mainly campesterol and sitosterol [25]. Finally, our study provides valuable data on physiological PS levels in a free-living European population.

There are several limitations to this study, pertaining to intrinsic characteristics of a cross-sectional observational study. First, normally occurring PS serum levels in this study fell within a narrow range as compared to variations achieved in interventional studies when using PS-enriched foods. Still, the values were comparable to those obtained in other countries, thus suggesting that the ranges observed might well fall within physiological values. Second, the assessment methods for IL-6 and TNF-α differed between studies, which might partly explain the observed inconsistencies; still, CRP levels were assessed in the same laboratory using the same method for both follow-ups, and inconsistencies were found. Third, the study was conducted in a geographically limited region, and it has been shown that dietary intakes vary between regions in Switzerland [46]. Hence, the results might not be replicable in other settings, and it would be interesting to conduct similar studies in other countries or locations. Fourth, we had limited information regarding other comorbidities susceptible to modify inflammatory status, such as cancer or chronic kidney disease; still, we believe that further adjusting for those comorbidities would decrease even more the number of associations between PS and inflammatory markers.

## 5. Conclusions

Our results failed to indicate a significant association between inflammatory markers and phytosterols levels in a community-dwelling Swiss population. Further studies should investigate the possibility of such an association under dietary conditions providing a higher phytosterol intake.

## Figures and Tables

**Figure 1 nutrients-14-02500-f001:**
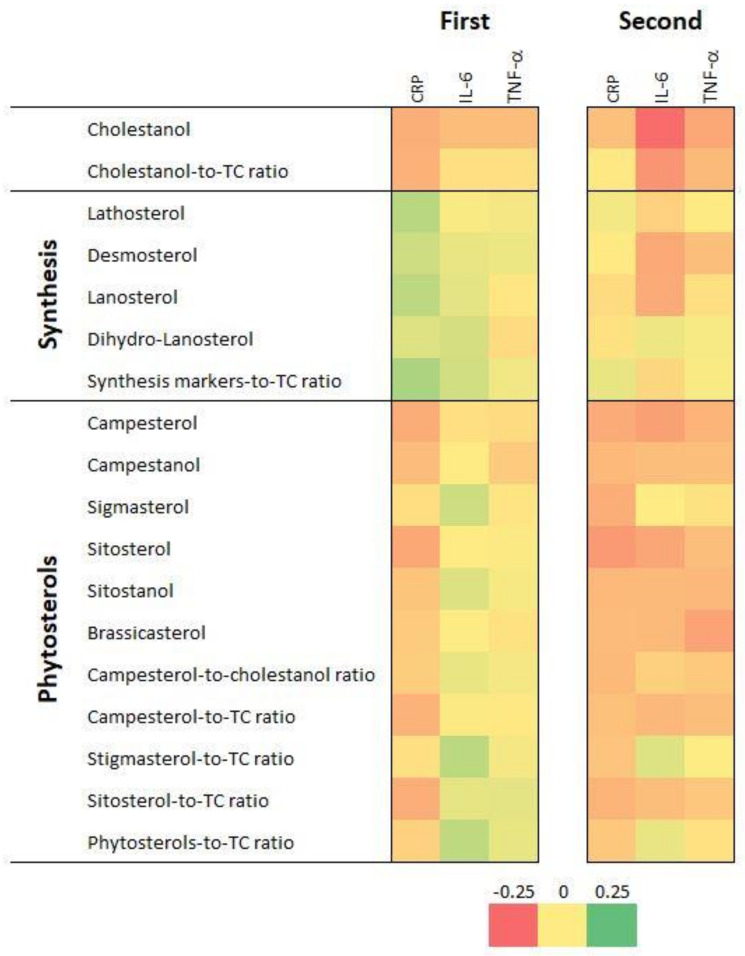
Bivariate correlation matrices between inflammatory markers and serum sterol levels and ratios, first (2009–2012) and second (2014–2017) follow-up, CoLaus|PsyCoLaus study, Lausanne, Switzerland. CRP, C-reactive protein; IL-6, interleukin 6; TNF-α, tumor necrosis factor alpha; TC, total cholesterol. Results for the first follow-up are in the first rectangle and for the second follow-up, in the second rectangle. Associations are represented on a color scale, going from red (strongly negative coefficients) to green (strongly positive coefficients), with coefficient values close to zero represented in yellow.

**Table 1 nutrients-14-02500-t001:** Sample characteristics, first (2009–2012) and second (2014–2017) follow-up, CoLaus|PsyCoLaus study, Lausanne, Switzerland.

	First (2009–2012)	Second (2014–2017)
Sample size	730	526
Women (%)	418 (57.3)	306 (58.2)
Age (years)	70.1 ± 4.7	75.0 ± 4.4
BMI (kg/m^2^)	26.5 ± 4.6	26.2 ± 4.4
BMI categories (%)		
Normal	295 (40.4)	225 (42.8)
Overweight	297 (40.7)	216 (41.1)
Obese	138 (18.9)	85 (16.2)
Smoking status (%)		
Never	293 (40.2)	215 (42.2)
Former	328 (45.0)	234 (45.9)
Current	108 (14.8)	61 (12.0)
Diabetes (%)	120 (16.5)	75 (14.3)
Treated with metformin (%)	46 (6.3)	50 (9.5)
Hypolipidemic drugs (%)		
Statins	159 (21.8)	128 (24.3)
Ezetimibe	6 (0.8)	6 (1.1)
Inflammatory markers		
CRP (mg/L)	1.7 [0.8–3.2]	1.7 [0.8–3.2]
IL-6 (ng/L)	2.75 [0.97–8.97]	2.75 [0.97–8.97]
TNF-α (ng/L)	4.84 [2.75–8.60]	4.84 [2.75–8.60]

BMI, body mass index; CRP, C-reactive protein, IL-6, interleukin 6; TNF-α, tumor necrosis factor alpha. Results are expressed as number of participants (column percentage) for categorical variables and as average ± standard deviation or as median [interquartile range] for continuous variables.

**Table 2 nutrients-14-02500-t002:** Serum sterol concentrations and sterol ratios, first (2009–2012) and second (2014–2017) follow-up, CoLaus|PsyCoLaus study, Lausanne, Switzerland.

	First (2009–2012)		Second (2014–2017)	
	Average ± SD	Median [IQR]	Average ± SD	Median [IQR]
Cholesterol absorption				
Cholestanol [mg/dL]	0.31 ± 0.10	0.30 [0.24–0.36]	0.41 ± 0.12	0.40 [0.33–0.48]
Cholesterol synthesis				
Lathosterol [mg/dL]	0.23 ± 0.12	0.21 [0.14–0.29]	0.20 ± 0.20	0.18 [0.11–0.25]
Desmosterol [mg/dL]	0.15 ± 0.15	0.13 [0.10–0.17]	0.16 ± 0.12	0.14 [0.10–0.19]
Lanosterol [µg/dL]	23.9 ± 14.7	21.1 [13.8–29.7]	18.8 ± 6.6	17.9 [14.1–22.5]
Dihydro-lanosterol [µg/dL]	3.95 ± 3.26	2.61 [1.91–4.35]	0.39 ± 0.30	0.34 [0.21–0.49]
Vegetal origin				
Campesterol [mg/dL]	0.33 ± 0.20	0.29 [0.19–0.42]	0.29 ± 0.16	0.26 [0.18–0.37]
Sitosterol [mg/dL]	0.25 ± 0.13	0.23 [0.16–0.31]	0.25 ± 0.12	0.23 [0.17–0.31]
Brassicasterol [µg/dL]	19.6 ± 10.9	17.5 [11.9–24.3]	20.7 ± 10.4	18.7 [13.9–24.8]
Sitostanol [µg/dL]	7.32 ± 4.05	6.45 [4.62–8.96]	4.18 ± 2.10	3.84 [3.31–4.52]
Campestanol [µg/dL]	6.08 ± 3.92	5.22 [3.39–7.75]	4.01 ± 1.50	3.75 [3.12–4.63]
Stigmasterol [µg/dL]	6.43 ± 3.97	5.41 [3.40–8.43]	7.96 ± 3.48	7.14 [5.65–9.31]
Ratios				
Cholestanol-to-TC ratio	1.38 ± 0.35	1.34 [1.13–1.58]	2.05 ± 0.52	2.00 [1.72–2.32]
Synthesis markers-to-TC ratio	106.5 ± 62.7	93.6 [63.8–128.5]	94.0 ± 30.2	88.0 [74.7–106.4]
Campesterol-to-cholestanol ratio	1.04 ± 0.53	0.96 [0.68–1.28]	0.72 ± 0.36	0.65 [0.47–0.87]
Campesterol-to-TC ratio (100×)	1.45 ± 0.86	1.30 [0.86–1.85]	1.47 ± 0.85	1.30 [0.90–1.80]
Stigmasterol-to-TC ratio	28.8 ± 17.6	24.6 [15.4–37.6]	40.1 ± 19.0	35.1 [27.4–47.1]
Sitosterol-to-TC ratio	1.14 ± 0.56	1.02 [0.75–1.38]	1.27 ± 0.67	1.12 [0.85–1.54]
Phytosterols-to-TC ratio	64.6 ± 33.3	56.4 [39.2–83.3]	64.2 ± 27.3	58.4 [46.8–73.4]

SD, standard deviation; IQR, interquartile range; TC, total cholesterol. Results are expressed as average ± standard deviation and median [interquartile range].

**Table 3 nutrients-14-02500-t003:** Robust regression between inflammatory markers and serum sterol levels and ratios, first (2009–2012) and second (2014–2017) follow-up, CoLaus|PsyCoLaus study, Lausanne, Switzerland.

	CRP		IL-6		TNF-α	
	FU1	FU2	FU1	FU2	FU1	FU2
**Cholesterol absorption**						
Cholestanol [mg/dL]	−0.554 (−1.584; 0.475)	−0.629 (−1.520; 0.262)	−0.609 (−3.281; 2.063)	**−0.554** **(−0.809; −0.300)**	**−3.375** **(−6.516; −0.234)**	**−1.284** **(−2.163; −0.405)**
**Cholesterol synthesis**						
Lathosterol [mg/dL]	**1.363** **(0.489; 2.236)**	−0.260 (−1.296; 0.777)	0.088 (−2.172; 2.348)	0.042 (−0.113; 0.196)	1.486 (−1.197; 4.168)	−0.014 (−0.541; 0.512)
Desmosterol [mg/dL]	−0.258 (−0.940; 0.424)	−0.189 (−0.999; 0.620)	−0.966 (−2.745; 0.812)	−0.230 (−0.471; 0.011)	0.339 (−1.741; 2.420)	−0.742 (−1.567; 0.083)
Lanosterol [µg/dL]	**0.009** **(0.002; 0.016)**	0.009 (−0.009; 0.027)	0.005 (−0.013; 0.023)	−0.004 (−0.010; 0.001)	0.003 (−0.018; 0.024)	0.005 (−0.014; 0.023)
Dihydro-lanosterol [µg/dl]	0.015 (−0.014; 0.045)	**0.410** **(0.046; 0.774)**	**0.111** **(0.035; 0.187)**	0.062 (−0.045; 0.168)	−0.031 (−0.121; 0.058)	0.186 (−0.179; 0.552)
**Vegetal origin**						
Campesterol [mg/dL]	0.155 (−0.351; 0.660)	−0.509 (−1.178; 0.160)	0.215 (−1.083; 1.513)	−0.140 (−0.337; 0.058)	−0.888 (−2.415; 0.639)	**−0.907** **(−1.573; −0.241)**
Sitosterol [mg/dL]	−0.121 (−0.901; 0.658)	−0.690 (−1.552; 0.172)	1.372 (−0.640; 3.385)	−0.182 (−0.438; 0.073)	−0.162 (−2.534; 2.210)	−0.714 (−1.585; 0.156)
Brassicasterol [µg/dL]	0.007 (−0.002; 0.016)	−0.008 (−0.018; 0.002)	0.013 (−0.009; 0.036)	−0.002 (−0.005; 0.001)	−0.004 (−0.031; 0.023)	**−0.019** **(−0.029; −0.010)**
Sitostanol [µg/dL]	−0.010 (−0.034; 0.014)	−0.007 (−0.055; 0.041)	0.037 (−0.024; 0.099)	−0.012 (−0.026; 0.003)	0.041 (−0.032; 0.114)	−0.043 (−0.091; 0.006)
Campestanol [µg/dL]	−0.013 (−0.038; 0.012)	−0.054 (−0.122; 0.014)	−0.001 (−0.064; 0.062)	−0.012 (−0.032; 0.009)	−0.036 (−0.111; 0.039)	−0.066 (−0.135; 0.002)
Stigmasterol [µg/dL]	0.009 (−0.016; 0.034)	−0.010 (−0.040; 0.020)	**0.104** **(0.040; 0.168)**	0.004 (−0.004; 0.013)	−0.039 (−0.113; 0.036)	0.011 (−0.019; 0.042)
**Ratios**						
Cholestanol-to-TC ratio	−0.135 (−0.415; 0.144)	−0.042 (−0.235; 0.151)	0.263 (−0.459; 0.985)	**−0.128** **(−0.183; −0.073)**	−0.580 (−1.437; 0.276)	**−0.389** **(−0.581; −0.198)**
Synthesis markers-to-TC ratio	**0.002** **(0.001; 0.004)**	**0.004** **(0.001; 0.008)**	0.003 (−0.001; 0.007)	−0.001 (−0.002; 0.000)	0.003 (−0.002; 0.007)	0.000 (−0.004; 0.004)
Campesterol-to-cholestanol ratio	0.048 (−0.135; 0.231)	−0.119 (−0.408; 0.171)	0.132 (−0.338; 0.602)	0.007 (−0.079; 0.093)	0.019 (−0.538; 0.576)	−0.278 (−0.570; 0.015)
Campesterol-to-TC ratio (100×)	0.046 (−0.070; 0.163)	−0.078 (−0.206; 0.049)	0.154 (−0.145; 0.453)	−0.029 (−0.066; 0.009)	−0.126 (−0.479; 0.227)	**−0.193** **(−0.320; −0.066)**
Stigmasterol-to-TC ratio	0.003 (−0.003; 0.008)	0.000 (−0.006; 0.006)	**0.032** **(0.017; 0.047)**	0.001 (−0.001; 0.002)	−0.003 (−0.020; 0.014)	0.001 (−0.005; 0.007)
Sitosterol-to-TC ratio	−0.024 (−0.205; 0.157)	−0.105 (−0.268; 0.058)	**0.510** **(0.049; 0.972)**	−0.038 (−0.087; 0.010)	0.183 (−0.367; 0.733)	**−0.168** **(−0.332; −0.003)**
Phytosterols-to-TC ratio	0.000 (−0.003; 0.003)	0.001 (−0.003; 0.005)	**0.013** **(0.005; 0.021)**	0.000 (−0.001; 0.001)	0.004 (−0.005; 0.013)	−0.002 (−0.006; 0.002)

CRP, C-reactive protein; FU, follow-up; IL-6, interleukin 6; TNF-α, tumor necrosis factor alpha. Results are expressed as slope and (95% confidence interval). Statistical analysis conducted by robust regression, adjusting for age (continuous), sex (male, female), BMI (continuous), diabetes (yes, no) and statin use (yes, no). Significant (*p* < 0.05) associations are indicated in bold.

## Data Availability

Due to the sensitivity of the data and the lack of consent for online posting, individual data cannot be made accessible. Only metadata will be made available in digital repositories. Metadata requests can also be performed via the study website www.colaus-psycolaus.ch.

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
