# Peer review of "Serum Phytosterols Are Not Associated with Inflammatory Markers in Two Cross-Sectional, Swiss Population-Based Studies (The CoLaus|PsyCoLaus Study)"

_nutrients, 2022, doi:10.3390/nu14122500_

Round 1
Reviewer 1 Report
The authors have prepared a clear and concise prospective clinical study investigating the association between serum phytosterol levels and inflammatory markers in a Swiss population. The particular strength in the designed study are two subsequent follow-up periods, which help solidify the findings. Additionally the data are strengthened by a clear and well-structured discussion. Even though the tested correlations do not show significance, the data presented have important implications for public health and is in line with other studies published in the field. The paper is could however be further improved with certain minor corrections.
11.) The authors showed insight, when correcting their analysis for statin use, however they could additionally use the reasoning why they decided to do so, either in introduction or discussion part. For example statin use has been associated with decreased levels of IL-6 and other inflammatory markers in the past.
22.) Figure 1 (heat map) is hard to understand and read. The text size could be bigger. Additionally not everything should be compared to one another. This is how you would gain more statistical power. The main changes are actually in the bolded brackets and are somehow overrun by the diagonal green, which is simply 1. The diagonal could be left blank (or crossed out) and then other significant and actual correlations would be more clear. Figure legend does not explain the marks in the quadrants and their meaning.
33.) The authors don´t mention other comorbidities such as kidney failure, diabetes… which could additionally contribute to low-grade inflammation and could influence serum inflammatory markers. Moreover some other drugs (for example metformin) alsohave an effect on serum IL-6 levels. These data should be added in a supplementary figure. If the data is not available, this should be listed as a clear limitation of the study.
44.) The discussion part on sterol levels (4.1) could be further improved if the authors included studies describing the contribution of genetics and gene polymorphisms on serum sterol levels. Since it was shown multiple times that diet as well as genetics play a role in this process. It would also further reinforce their rationalization regarding different sterol levels in different populations (Amish vs European).
Minor:
1- Write the definition of the abbreviation, when first used and then continue with the abbreviated form throughout the text for consistency reasons. For example PS – phytosterol, FU –follow up.
2- TNF-α is sometimes missing the α (line 213, 233, 269…)
3- Missing reference labelled as (ref) in line 224
4- Table 2 is very ´crowded. It would improve the readability if headings were bolded (e.g. ratios, vegetal origin…)
Author Response
Please see the attachment. We cannot upload the amended version of the manuscript due to the system's policy.

Reviewer 2 Report
In manuscript “Serum phytosterols are not associated with inflammatory markers in two cross-sectional, Swiss population-based studies. The CoLaus | PsyCoLaus study”, the authors measured the serum level of six PS, four cholesterol synthesis markers, one cholesterol absorption marker and some inflammatory markers. Although the result failed to indicate a significant association between inflammatory markers and phytosterols level in the serum samples, the work is still significant and provides useful information for future study. The manuscript is well written and the data are presented in a logical way.
